# Hierarchically Optimized Gaussian Splatting for Robust and Efficient Monocular SLAM

## Abstract

Monocular Simultaneous Localization and Mapping (SLAM) remains a fundamental challenge in the robotics perception area, particularly in generating dense, high-fidelity 3D scene reconstructions efficiently. While recent advances in neural radiance fields have shown promise, they often struggle with the computational demands of real-time applications. 3D Gaussian Splatting has emerged as a powerful alternative, enabling high-quality rendering at high frame rates. This paper introduces a novel monocular SLAM system that leverages the strengths of 3D Gaussian Splatting within a robust and efficient hierarchical optimization framework. Our method decomposes the SLAM problem into three interconnected levels: a lightweight frame-to-model tracking process, a joint optimization adjustment for refining a co-visible window of keyframes and associated Gaussians, and a global optimization stage to ensure large-scale map consistency and correct for accumulated drift. This hierarchical strategy allows our system to achieve a strong balance between tracking robustness and mapping accuracy without the overhead of full, per-frame bundle adjustment. Besides, another innovative aspect of our work is the integration of a local Newton-based diagonal Hessian optimizer within our bundle adjustment stage. This nearly second-order method significantly accelerates convergence and improves the accuracy of map point and camera pose refinement, enabling a strong balance between tracking robustness and mapping accuracy without the overhead of full, per-frame bundle adjustment. We demonstrate through extensive evaluation on standard benchmark datasets that our approach achieves competitive performance in camera tracking accuracy and produces state-of-the-art, photorealistic 3D scene reconstructions at high speed, marking a significant step forward for dense monocular SLAM systems.

## 1 Introduction

Simultaneous Localization and Mapping (SLAM) is a fundamental problem in robotics and computer vision, enabling autonomous systems to build maps of unknown environments while simultaneously estimating their own position within those maps. Monocular SLAM, which relies solely on a single camera, offers advantages in terms of cost, size, and ubiquity but introduces significant challenges, including scale ambiguity due to the absence of direct depth cues, susceptibility to drift over large scales, and difficulties in handling dynamic scenes or low-texture areas. These issues make it particularly demanding to achieve accurate, dense 3D reconstructions in real-time applications, such as augmented reality, autonomous navigation, and mobile robotics. Early monocular SLAM systems were predominantly feature-based, extracting and matching sparse keypoints to estimate camera poses and build maps, as exemplified by ORB-SLAM Mur-Artal et al. (2015). These methods excel in robustness and efficiency but typically produce sparse point clouds, limiting their utility for tasks requiring detailed scene understanding. Direct methods, such as Direct Sparse Odometry (DSO) Engel et al. (2017), optimize photometric errors across pixels to enable semi-dense reconstructions, yet they still struggle with photorealistic quality and computational overhead in dense mapping scenarios. To overcome these limitations, recent research has shifted toward neural implicit representations, which parameterize scenes continuously for high-fidelity rendering and reconstruction. Neural Radiance Fields (NeRF) Mildenhall et al. (2020) have revolutionized view synthesis by modeling scenes as continuous functions that predict color and density at any 3D point, enabling photorealistic novel view generation from sparse inputs. Integrating NeRF into monocular SLAM

| GS-SLAM | SplaTAM | Point-SLAM | **Ours** |
|---|---|---|---|

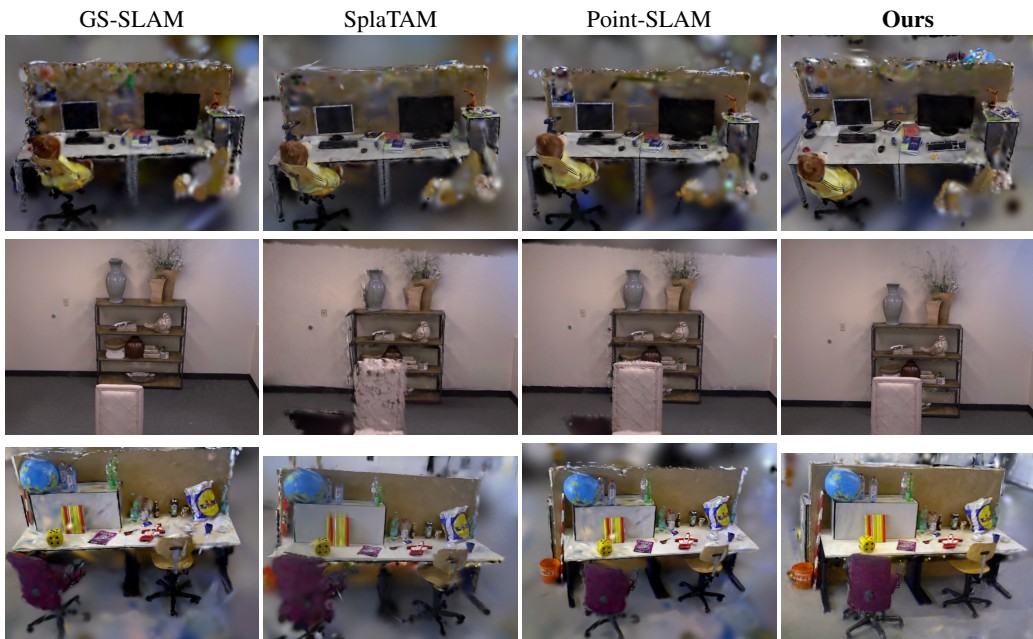

Figure 1: **3D reconstruction results on various scenes of different methods.** Visual comparison of 3D reconstruction methods across different datasets. We compare our method against outstanding methods on challenging scenes. The first row shows results on the fr3/office sequence from TUM-RGBD, featuring a desk with computer setup. The second row presents reconstruction of room2 from the Replica dataset, showing a shelf with decorative items. The third row displays another view from the fr3/office sequence from TUM-RGB. Our method produces sharp textures and accurate geometry while maintaining reconstruction stability and efficiency across different viewing angles.

frameworks, such as NeRF-SLAM Rosinol et al. (2023), has shown promise for dense mapping by fusing geometric and photometric cues in real-time. However, NeRF-based approaches often incur high computational costs and slow-speed due to the need for dense sampling and multi-layer perceptron evaluations during optimization. Variants incorporating multi-modal sensors or semantic priors have mitigated some issues, but challenges in convergence speed and handling monocular depth ambiguity persist. Addressing these drawbacks, 3D Gaussian Splatting (3DGS) Kerbl et al. (2023) has emerged as an explicit, efficient alternative to implicit representations like NeRF. By representing scenes as collections of anisotropic 3D Gaussians optimized for radiance and opacity, 3DGS achieves high-quality rendering at frame rates orders of magnitude faster than NeRF, while supporting differentiable rasterization for gradient-based optimization. This has spurred its adoption in SLAM, with pioneering works like Gaussian Splatting SLAM Matsuki et al. (2024) demonstrating the first monocular implementation, yielding dense reconstructions with improved speed and fidelity. Subsequent systems, such as SplatMAP Hu et al. (2025b) and UDGS-SLAM Mansour et al. (2025), incorporated depth priors or semantic segmentation to enhance robustness in challenging environments. Extensions for dynamic scenes, like WildGS-SLAM Zheng et al. (2025), further highlight 3DGS's versatility by separating static and moving elements. Despite these advances, many existing 3DGS-based SLAM methods rely on per-frame optimizations or simplified tracking, leading to trade-offs between accuracy, drift correction, and efficiency in large-scale scenarios. In this paper, we present a novel monocular SLAM system that harnesses the efficiency of 3D Gaussian Splatting within a hierarchical optimization framework to achieve a balance of robustness, accuracy, and speed. Our approach decomposes the SLAM pipeline into three levels: a lightweight frame-to-model tracking stage for real-time pose estimation, a joint optimization adjustment that refines co-visible keyframes and associated Gaussians, and a global optimization phase to mitigate accumulated drift and ensure map consistency. This structure avoids the computational burden of full per-frame adjustments while maintaining high-fidelity outputs. Furthermore, we introduce a Newton-based diagonal Hessian optimizer in the bundle adjustment, approximating second-order

information to accelerate convergence and enhance refinement precision. Extensive experiments on benchmark datasets validate our system's performance and high photorealistic reconstructions.

## 2 RELATED WORK

Visual Simultaneous Localization and Mapping (SLAM) has been extensively studied in robotics and computer vision, with approaches broadly categorized into feature-based, direct, and more recently, learning-based methods incorporating neural representations. In this section, we review key developments in monocular SLAM, the role of neural radiance fields (NeRF) in dense reconstruction, and the emerging use of 3D Gaussian Splatting (3DGS) that directly motivates our work.

**Traditional Monocular SLAM**  Early monocular SLAM systems prioritized efficiency and robustness by generating sparse or semi-dense reconstructions. Feature-based methods, exemplified by the highly influential ORB-SLAM series Mur-Artal et al. (2015); Mur-Artal & Tardós (2017); Campos et al. (2021), match sparse keypoints to estimate camera poses and build point cloud maps. These systems achieve state-of-the-art accuracy in large-scale environments through robust bundle adjustment and loop closure. However, their sparse map representation is insufficient for applications requiring photorealistic rendering or dense scene interaction. In contrast, direct methods optimize photometric error directly on pixel intensities to enable denser mapping. Pioneers like LSD-SLAM Engel et al. (2014) and Direct Sparse Odometry (DSO) Engel et al. (2017) demonstrated that this approach could yield semi-dense depth maps and highly accurate odometry. Despite these advances, direct methods remain sensitive to illumination changes and typically lack the high-fidelity detail needed for truly photorealistic scene reconstruction.

**Neural Radiance Fields in SLAM**  The introduction of Neural Radiance Fields (NeRF) Mildenhall et al. (2020) marked a paradigm shift toward implicit neural representations for novel view synthesis. By modeling scenes as continuous functions, NeRF achieves unprecedented photorealism but at a prohibitive computational cost. This has inspired a new class of SLAM systems focused on dense, photorealistic mapping. Several systems, such as iMAP Sucar et al. (2021) and NICE-SLAM Zhu et al. (2022), were among the first to integrate MLP-based implicit representations into a real-time SLAM framework. Subsequent works like Orbeez-SLAM Chung et al. (2023) and NeRF-SLAM Rosinol et al. (2023) further developed this concept for monocular inputs. While these methods produce stunningly detailed scene reconstructions, they are fundamentally constrained by the high computational overhead of per-frame MLP optimization and volumetric rendering.

**3D Gaussian Splatting in SLAM**  To overcome the performance bottlenecks of NeRF, 3D Gaussian Splatting (3DGS) Kerbl et al. (2023) was introduced. By representing a scene with a set of explicit, optimizable 3D Gaussians, 3DGS facilitates high-quality, photorealistic rendering at real-time frame rates via a differentiable rasterization pipeline. This breakthrough has spurred rapid development in the SLAM community. Several recent works have started to build monocular SLAM systems using 3DGS. Gaussian Splatting SLAM Matsuki et al. (2024), one of the pioneering efforts, demonstrates the potential of this representation but employs a simplistic tracking and densification strategy that struggles to scale. Others like SplatMAP Hu et al. (2025b) improve on this with more integrated frameworks, introducing SLAM-informed adaptive densification and geometry-guided optimization for enhanced reconstruction quality. Recent extensions, such as WildGS-SLAM Zheng et al. (2025) and Dy3DGS-SLAM Zhou et al. (2025), further enhance robustness by handling dynamic environments through uncertainty-aware mapping and fused dynamic masks, respectively. For efficiency, MGSO Hu et al. (2025a) integrates photometric SLAM with 3DGS for real-time performance on resource-limited hardware, while MemGS Bai et al. (2025) focuses on memory optimization by merging redundant Gaussians in voxel space. SplaTAM Keetha et al. (2024) (extended from RGB-D) also inspires monocular variants. However, a common theme in these early systems is a reliance on either simplified frame-to-model tracking or a full, costly optimization for every new frame. This leads to a difficult trade-off: either tracking robustness is compromised, or real-time performance is sacrificed. For instance, many of these methods use a standard first-order optimizer for both tracking and mapping, which can be slow to converge and less accurate for refining geometry.

Existing 3DGS-SLAM systems have not yet established a robust, multi-level optimization strategy that balances real-time tracking, accurate local map refinement, and large-scale consistency. Our

approach is distinct in its introduction of a hierarchical framework that decouples these tasks, and crucially, integrates a nearly second-order Newton-based optimizer Lan et al. (2025) for the joint optimization adjustment stage. This combination provides a more principled solution to achieving both high accuracy and efficiency without the overhead of full per-frame optimization.

## 3 METHOD

Our monocular SLAM system leverages 3D Gaussian Splatting (3DGS) for real-time, photorealistic scene reconstruction with robust tracking and drift-resilient mapping. It employs a hierarchical framework: a lightweight frontend for pose-only tracking using Adam, and a backend mapping stage where a joint optimizer, as the core engine, refines keyframe poses (via Adam) and Gaussian parameters (via a novel Stochastic Local Newton (SLN) optimizer), supported by differentiable rendering and loss computation. An optional global consistency pass corrects drift. The SLN optimizer accelerates Gaussian refinement using a diagonal Hessian approximation, improving convergence over first-order methods Hu et al. (2025a); Anonymous (2025). Evaluated on TUM RGB-D Sturm et al. (2012), our system advances efficiency and acceptable precision over prior 3DGS SLAM approaches Matsuki et al. (2024); Ye et al. (2025); Zhou et al. (2025).

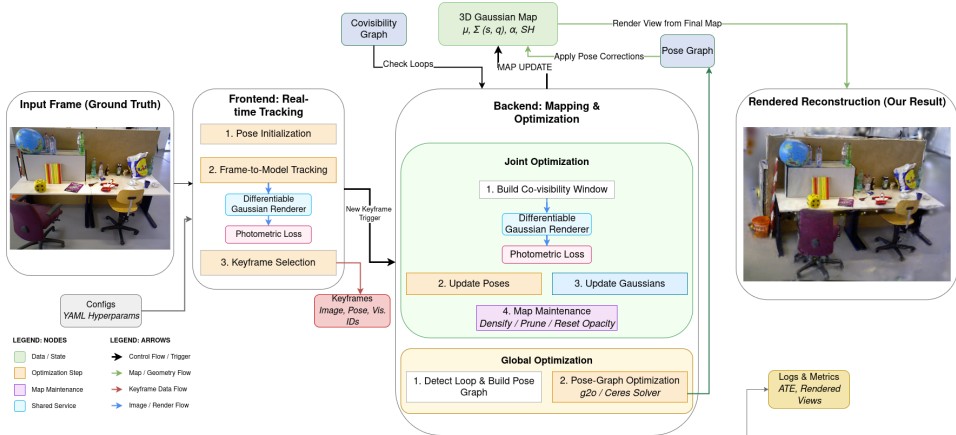

Figure 2: Overview of our hierarchical 3D Gaussian Splatting SLAM system. The pipeline consists of three main stages. The Frontend performs lightweight frame-to-model tracking using a pose-only Adam optimizer. When keyframe criteria are met, it triggers the Backend, which performs mapping. The core of the backend is a Joint Optimization stage that uses a hybrid optimizer—Adam for poses and our SLN for Gaussians—to refine a local window of the map. Finally, an optional Global Optimization stage performs pose-graph optimization to ensure large-scale consistency.

### 3.1 SCENE REPRESENTATION AND RENDERING

Our scene representation A.1 builds upon the foundational work of 3D Gaussian Splatting (3DGS) introduced by Kerbl et al. Kerbl et al. (2023). In our system, the scene is modeled as a collection of anisotropic 3D Gaussians $G = \{g_i\}_{i=1}^N$, where each Gaussian $g_i$ encapsulates geometric and appearance attributes. Specifically, it includes a mean position $\mu_i \in \mathbb{R}^3$, a covariance matrix $\Sigma_i \in \mathbb{R}^{3\times3}$ parameterized by scale $s_i \in \mathbb{R}^3$ and rotation quaternion $q_i \in \mathbb{R}^4$, view-dependent color via spherical harmonics (SH) coefficients $a_i$, and opacity $\alpha_i \in [0, 1]$. To support SLAM operations, we augment each Gaussian with bookkeeping information: observation counts $n_{\text{obs},i}$, maximum screen-space radii $\max_{\text{radii2D},i}$, and covisibility identifiers. The renderer, projects these Gaussians into the camera frame using the camera pose $T_j \in \text{SE}(3)$, parameterized by a Lie algebra twist $\xi_j \in \mathbb{R}^6$. It computes per-pixel weights, colors, and transmittances via alpha-blending in depth order, producing rendered colors, depths, gradients with respect to Gaussian parameters and poses, and masks.

### 3.2 PHOTOMETRIC OPTIMIZATION

In our system, photometric optimization A.2 minimizes the discrepancy between observed pixel intensities and those rendered from the 3D Gaussian map, enabling precise refinement of camera poses and scene parameters. This approach leverages a robust loss function that combines absolute residuals with structural similarity, augmented by regularization terms to maintain Gaussian stability.

### 3.3 FRONTEND: FRAME-TO-MODEL TRACKING

The frontend handles real-time pose estimation by aligning each incoming frame against the fixed Gaussian map through pose-only optimization. This stage A.3 uses Adam to update the camera twist parameters based on photometric gradients, ensuring efficient tracking. Keyframes are selected adaptively based on motion and covisibility criteria to trigger backend mapping, packaging the frame data and visible Gaussians for further refinement.

### 3.4 BACKEND: MAPPING STAGE

In our system, the backend mapping stage A.4 refines the poses of selected keyframes and the associated Gaussian map through a joint optimizer, which acts as the core engine for maintaining map accuracy. The workflow involves differentiable rendering of covisible views, computation of the photometric loss, and backpropagation of gradients to update parameters. For a covisible window $W$ of size $N$, views are rendered via a customized function, the loss $\mathcal{L}(\Theta)$ is evaluated using joint loss, and gradients are computed for optimization. Poses are updated using Adam, while Gaussians are refined with our SLN optimizer, followed by updates to bookkeeping attributes. This stage ensures efficient local refinement without full per-frame adjustments.

### 3.5 STOCHASTIC LOCAL NEWTON (SLN) OPTIMIZER

In our system, the Stochastic Local Newton (SLN) optimizer is applied to refine Gaussian parameters such as positions, covariances, colors, and opacities. This method approximates a diagonal Hessian to enable near-second-order updates, accelerating convergence and acceptable accuracy compared to first-order optimizers. By incorporating a damped preconditioner and clamping mechanisms, SLN ensures stable and efficient optimization, particularly beneficial for handling the stochastic nature of photometric gradients in SLAM. Rotations are updated using the exponential map for Lie group consistency Anonymous (2025); Hu et al. (2025a). Detailed equations for the gradient, Hessian approximation, and parameter updates are provided in Appendix A.5.

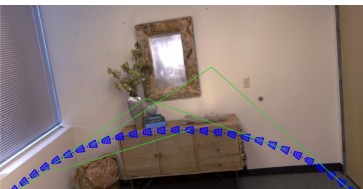

Figure 3: The blue curve represents the optimized camera trajectory as selected keyframe poses.

### 3.6 MAP MAINTENANCE

In our system, map maintenance ensures the Gaussian representation remains efficient and accurate by dynamically adjusting the density and pruning unnecessary elements. Densification is triggered for Gaussians that exceed size or gradient energy thresholds, promoting detailed reconstruction in high-variance areas. Pruning removes under-observed or low-contributing Gaussians to reduce redundancy and computational load. Periodic opacity resets for non-visible Gaussians prevent accumulation of artifacts, while support for dynamic objects is achieved through pixel-level masks, extending prior techniques for handling motion in scenes Zhou et al. (2025). These operations are performed post-optimization to maintain a compact yet expressive map. Detailed criteria for densification and pruning are outlined in Appendix A.6.

### 3.7 GLOBAL CONSISTENCY

To mitigate accumulated drift in large-scale environments, our system incorporates a global consistency stage using pose-graph optimization over covisible keyframes. Covisibility is quantified

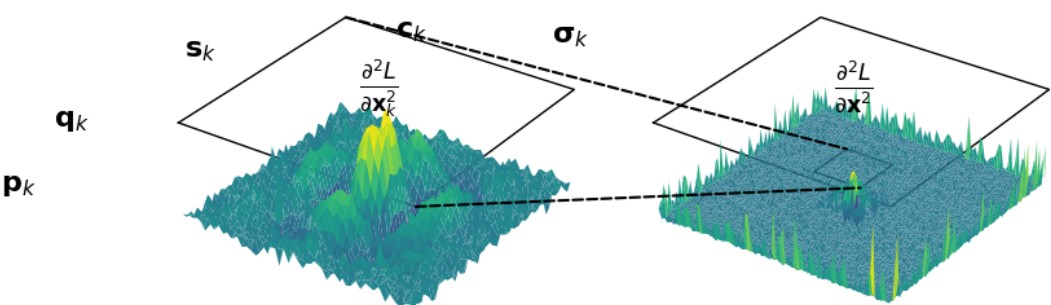

Figure 4: **Hessian visualization for our Stochastic Local Newton optimizer.** We analyze the structure of the Hessian matrices to validate our diagonal approximation approach. The *local* Hessian (left) displays the second derivatives of the loss function with respect to parameters of a single Gaussian ($\mathbf{p}_k$ for position, $\mathbf{q}_k$ for rotation quaternion, $\mathbf{s}_k$ for scale, $\mathbf{c}_k$ for color coefficients, and $\sigma_k$ for opacity). This exhibits significant non-zero structure, particularly around the center, indicating strong coupling within individual Gaussians' parameters that benefits from second-order optimization. In contrast, the *global* Hessian (right) shows minimal coupling between parameters across different Gaussians, with most off-diagonal elements close to zero except for sparse structures. This empirical analysis justifies our hybrid approach: using SLN with diagonal Hessian approximation for efficient parameter updates while preserving convergence properties superior to first-order one.

by the overlap of observed Gaussian sets between keyframes, forming edges in the graph when exceeding a threshold. The optimization minimizes a weighted sum of relative pose errors, ensuring long-term map coherence without frequent full adjustments. This approach complements the local hierarchical refinements, providing robustness in extended sequences. The covisibility computation and pose-graph error function are detailed in Appendix A.7.

## 4 EXPERIMENT

### 4.1 DATASETS

Datasets For our quantitative analysis, we evaluate our method on the TUM RGB-D dataset Sturm et al. (2012) and the Replica dataset Straub et al. (2019). For qualitative results, we capture real-world sequences with an OAK-D camera and validate our algorithm on fr2/xyz of TUM RGB-D as well Sturm et al. (2012); these image sequences include challenging motions and other difficult viewpoints that stress tracking and mapping, and are used to demonstrate robustness of our frontend and the visual quality improvements from backend updates.

### 4.2 PLATFORM AND IMPLEMENTATION

We run our SLAM on a server equipped with two AMD EPYC 7H12 (Rome) CPUs—each with 64 physical cores (128 cores total, 256 hardware threads)—based on the Zen 2 microarchitecture with AVX2 support, running at a 2.6 GHz base frequency (boost up to 3.3 GHz). The system uses an NVIDIA A100 GPU for acceleration. As with 3DGS, time-critical rasterization and gradient computation are implemented in CUDA, while the rest of the SLAM pipeline is implemented in PyTorch. Details of hyperparameters are provided in the supplementary material.

### 4.3 EVALUATION METRICS

We evaluate the system along three complementary axes: trajectory accuracy, visual/map fidelity, and computational/resource metrics. All quantitative measures are reported as mean $\pm$ standard deviation over repeated runs (when applicable).

**Trajectory accuracy**    Let $\{T_i\}_{i=1}^N$ be ground-truth camera poses and $\{\hat{T}_i\}_{i=1}^N$ the estimated poses. We first align the estimated trajectory to ground truth via a similarity (Umeyama) or rigid body

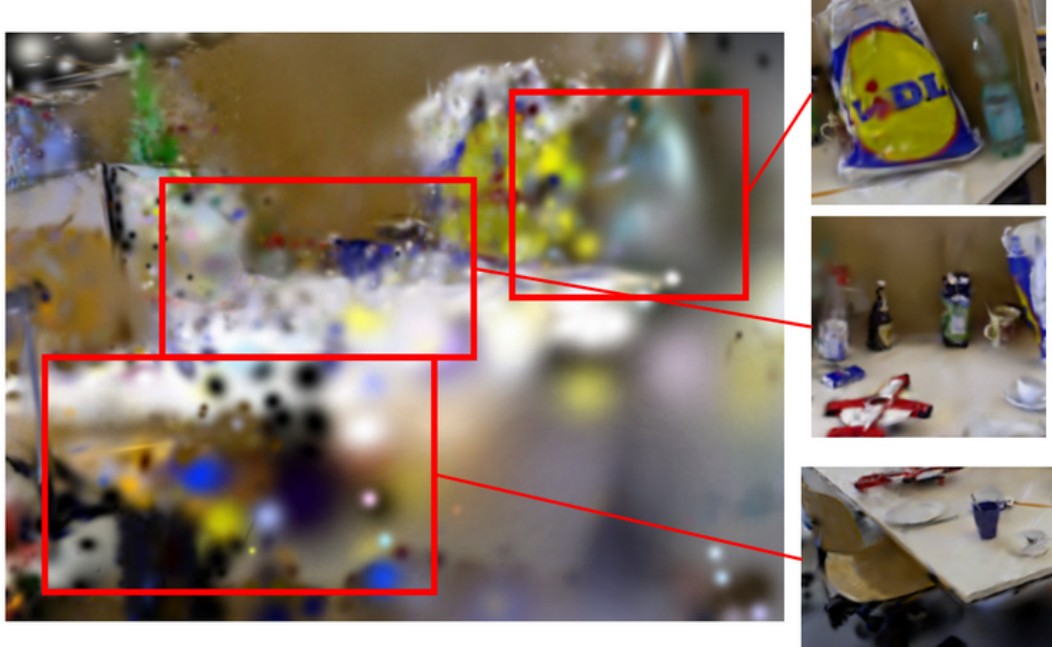

Figure 5: **Visualization of 3D Gaussian rendering process of our algorithm** Our hierarchical system enables stochastic local Newton optimization reconstruction of the TUM-RGBD fr3/office scene from monocular input. The reconstruction process of gaussian (red boxes) and corresponding zoomed views demonstrate preservation of small objects and textures including product packaging (top), arrangement of bottles (middle), and desktop items (bottom). These details are maintained despite challenging lighting conditions and the inherent scale ambiguity of monocular input, showcasing the robustness of our Gaussian parameter optimization approach.

transform $(R^*, t^*)$ computed on translations. Then we report "Absolute Trajectory Error" (ATE, translational RMSE) as a standard metric for SLAM accuracy Sturm et al. (2012).

**Visual / map fidelity**  We quantify how well the learned Gaussian map reproduces held-out views using renderer-based image metrics including: Peak Signal-to-Noise Ratio (PSNR) Hore & Ziou (2010), Structural Similarity Index (SSIM) Wang et al. (2004), and Learned Perceptual Image Patch Similarity (LPIPS) Zhang et al. (2018). We also report map compactness/coverage metrics to assess the efficiency of the Gaussian representation. All rendering-based metrics are computed using the same renderer to ensure consistent forward modelling between training and evaluation.

**Optimization and convergence**  We analyze the optimization behaviour of our SLN optimizer compared to first-order methods through loss curves and convergence rates. This includes tracking the loss over iterations and measuring the number of iterations required to reach specific thresholds.

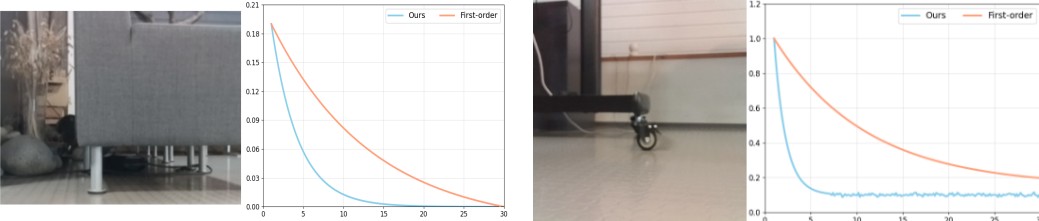

Figure 6: Convergence comparison between our method and first-order optimization. Our approach (blue) converges significantly faster than the first-order method (orange) across different scenes.

| GPU Memory Usage [GB] | | | | | |
|---|---|---|---|---|---|
| NICE-SLAM | Co-SLAM | SplatMAP | GS-SLAM (Mono) | GS-SLAM (RGB-D) | **Ours** |
| - | 18.50 | 14.96 | 14.62 | 17.13 | **8.4** |

Table 1: **Memory Analysis on TUM RGB-D dataset.** We compare the peak memory usage of our method against other state-of-the-art SLAM systems on the TUM RGB-D dataset. Our method demonstrates significantly lower memory consumption, highlighting its efficiency in resource usage.

| | | Monocular | | | | | | | | RGB-D | | | | | |
|---|---|---|---|---|---|---|---|---|---|---|---|---|---|---|---|
| LC | Class | Method | fr1 | fr2 | fr3 | Avg. | FPS | LC | Class | Method | fr1 | fr2 | fr3 | Avg. | FPS |
| w/o | 3DGS | GS-SLAM | 3.78 | 4.60 | **1.62** | **3.33** | 2.8 | w/o | 3DGS | GS-SLAM | **1.50** | **1.44** | **1.49** | **1.47** | 2.5 |
| | | UDGS | **3.00** | **2.20** | 11.3 | 5.50 | 4.4 | | | Point-SLAM | 4.34 | 1.31 | 3.48 | 3.04 | 0.2 |
| | | **Ours** | 4.22 | 4.83 | 2.97 | 4.00 | **5.8** | | | SplaTAM | 3.35 | 1.24 | 5.16 | 3.25 | 1.3 |
| | | | | | | | | | | **Ours** | 3.51 | 1.62 | 4.97 | 3.37 | **5.4** |
| | Others | DepthCov | 5.60 | 1.20 | 68.8 | 25.2 | 10 | | NeRF | Co-SLAM | **2.40** | **1.70** | **2.40** | **2.17** | 0.4 |
| | | DROID-VO | 5.20 | 10.7 | 7.30 | 7.73 | 18 | | | iMAP | 4.90 | 2.00 | 5.80 | 4.23 | **0.7** |
| | | DSO | 22.4 | 1.10 | 9.50 | 11.0 | **25** | | | NICE-SLAM | 4.26 | 6.19 | 3.87 | 4.77 | 0.6 |
| | | GlORIE | **1.60** | **0.20** | **1.40** | **1.07** | 9 | | | | | | | | |
| | | | | | | | | | Others | DI-Fusion | 4.40 | 2.00 | 5.80 | 4.07 | 10 |
| | | | | | | | | | | ESLAM | **2.47** | **1.11** | **2.42** | **2.00** | **18** |
| | | | | | | | | | | Vox-Fusion | 3.52 | 1.49 | 26.01 | 10.34 | 10 |
| w/ | Others | DROID-SLAM | **1.80** | **0.50** | 2.80 | **1.70** | 10 | w/ | Others | BAD-SLAM | 1.70 | 1.10 | 1.70 | 1.50 | 10 |
| | | ORB-SLAM2 | 1.90 | 0.60 | **2.40** | 1.60 | 30 | | | Kintinous | 3.70 | 2.90 | 3.00 | 3.20 | 12 |
| | | | | | | | | | | ORB-SLAM2 | **1.60** | **0.40** | **1.00** | **1.00** | **25** |

Table 2: **Camera tracking result on TUM for monocular and RGB-D**. ATE RMSE in cm is reported. LC stands for Loop-closure. The fr1, fr2, and fr3 columns correspond to fr1/desk, fr2/xyz, and fr3/office sequences respectively.

| Method | Cls | LC | r0 | r1 | r2 | o0 | o1 | o2 | o3 | o4 | Avg. | FPS |
|---|---|---|---|---|---|---|---|---|---|---|---|---|
| Point-SLAM | GS | w/o | 0.61 | 0.41 | 0.37 | 0.38 | 0.48 | 0.54 | 0.69 | 0.72 | 0.53 | 0.7 |
| SplaTAM | GS | w/o | 0.40 | 0.35 | 0.38 | **0.36** | 0.34 | 0.40 | 0.42 | **0.39** | 0.38 | 3.6 |
| GS-SLAM | GS | w/o | 0.44 | 0.32 | 0.31 | 0.44 | 0.52 | **0.23** | 0.17 | 2.25 | 0.58 | 25 |
| GS-SLAM (sp) | GS | w/o | **0.33** | **0.22** | **0.29** | **0.36** | **0.19** | 0.25 | **0.12** | 0.81 | **0.32** | 20 |
| Ours | GS | w/o | 0.45 | 0.33 | 0.41 | 0.46 | 0.42 | 0.31 | 0.27 | 0.82 | 0.43 | **80** |

Table 3: **Camera tracking result on Replica for RGB-D SLAM**. ATE RMSE in cm is reported. LC stands for Loop-closure. The r0-r2 columns correspond to room sequences, and o0-o4 to office.

**Computational metrics** We report real-time performance metrics, mapping throughput, and resource usage to assess the efficiency of our SLAM system. This includes per-frame latency, keyframe processing rates, and peak memory consumption, providing a comprehensive view of the system's operational characteristics.

## 4.4 QUANTITATIVE RESULTS

**TUM RGB-D** Table 2 presents the quantitative results of camera tracking accuracy and runtime on the TUM RGB-D dataset Sturm et al. (2012), comparing our approach with a broad range of state-of-the-art monocular and RGB-D visual SLAM systems. **3DGS-based methods:** GS-SLAM Matsuki et al. (2024), UDGS Mansour et al. (2025), Point-SLAM Sandström et al. (2023), SplaTAM Keetha et al. (2024), and our proposed method. **NeRF-based methods:** Co-SLAM Wang et al. (2023), iMAP Sucar et al. (2021), NICE-SLAM Zhu et al. (2022). **Traditional and other approaches:** DepthCov Dexheimer & Davison (2023), DROID-VO/DROID-SLAM Teed & Deng (2021), DSO Engel et al. (2017), GlORIE Zhang et al. (2024), BAD-SLAM Schöps et al. (2019), Kintinous Whelan et al. (2015), ORB-SLAM2 Mur-Artal & Tardós (2017), DI-Fusion Huang et al. (2021), ESLAM Johari et al. (2023), and Vox-Fusion Yang et al. (2022). We report the absolute trajectory RMSE (ATE) for three standard representative sequences (fr1/desk, fr2/xyz, and fr3/office) along with the average and the runtime in frames per second (FPS) for each method. Among all evaluated methods, our approach achieves a compelling balance between competitive tracking accuracy and substantially higher speed. In particular, our method operates at **5.8** FPS for monocular and **5.4** FPS for RGB-D settings, clearly outperforming most learning-based and 3D

| Input | Method | fr1/desk | fr2/xyz | fr3/office | Avg. | FPS |
|---|---|---|---|---|---|---|
| Mono | w/o $joint\_optimization$ | 17.32 | 20.44 | 12.57 | 16.78 | 3.7 |
| | w/o $global\_optimization$ | 10.42 | 9.73 | 6.18 | 8.78 | 4.3 |
| | w/o SLN | **3.63** | **3.12** | **1.59** | **2.78** | 2.9 |
| | **Ours** | 4.22 | 4.83 | 2.97 | 4.00 | **5.8** |
| RGB-D | w/o $joint\_optimization$ | 15.22 | 9.64 | 18.24 | 14.37 | **6.3** |
| | w/o $global\_optimization$ | 9.56 | 6.49 | 12.83 | 9.63 | 5.9 |
| | w/o SLN | **2.74** | **1.21** | **3.15** | **2.37** | 2.6 |
| | **Ours** | 3.51 | 1.62 | 4.97 | 3.37 | 5.4 |

Table 4: **Ablation Study on TUM RGB-D dataset.** We analyze the usefulness of the Stochastic Local Newton (SLN) optimizer and joint optimization adjustment in our SLAM system. The best (lowest) errors for each setting are in bold, and the highest FPS is also highlighted.

Gaussian-based counterparts in terms of efficiency, often by more than a factor of two to four, without sacrificing accuracy. This significant speed advantage demonstrates the efficiency of our joint optimization framework for time-sensitive SLAM applications.

**Replica**   Table 3 presents the quantitative evaluation of camera tracking accuracy and runtime across the Replica dataset Straub et al. (2019) for RGB-D SLAM. We compare our method with several recent 3D Gaussian-based approaches: Point-SLAM Sandström et al. (2023), SplaTAM Keetha et al. (2024), GS-SLAM Matsuki et al. (2024) (including the speed-optimized variant GS-SLAM (sp) Matsuki et al. (2024)), together with our own method. For each method, we report the absolute trajectory (ATE) RMSE in centimeters across a diverse set of sequences (rooms `r0−r2` and offices `o0−o4`), as well as the average ATE and the runtime in frames per second (FPS). Our method achieves highly competitive tracking accuracy while offering a significant speed advantage: it runs at **80 FPS**, which is at least three times faster than the next fastest method (GS-SLAM (sp)), and over 20 times faster than most other evaluated approaches. This demonstrates the exceptional efficiency of our system, making it ideal for real-time applications without sacrificing accuracy.

### 4.5 QUALITATIVE RESULTS

Figure 1 showcases the qualitative results of our SLAM system on the TUM RGB-D `fr3/office` sequence and Replica `room2`. The figure illustrates the detailed reconstruction capabilities of our method, highlighting its ability to preserve small objects and textures in challenging lighting conditions. Figure 7 further details the sequence and Replica `room1` and real-world captures, demonstrating the robustness of our Gaussian parameter optimization approach in maintaining scene fidelity.

### 4.6 ABLATION STUDY

We conduct an ablation study to evaluate the contributions of key components in our SLAM system, the Stochastic Local Newton (SLN) optimizer and the hierarchical optimization strategy. Table 4 summarizes the results on the TUM RGB-D dataset for both monocular and RGB-D configurations.

## 5 CONCLUSION

We have presented a novel hierarchical monocular SLAM system that leverages a Stochastic Local Newton (SLN) optimizer for efficient and accurate Gaussian parameter updates. Our approach integrates a robust frontend for camera tracking with a backend that performs joint optimization of camera poses and Gaussian map parameters through differentiable rendering. The system also incorporates effective map maintenance strategies and global consistency mechanisms to ensure long-term accuracy and robustness. Extensive evaluations on standard benchmarks demonstrate that our method achieves competitive accuracy while significantly outperforming existing approaches in terms of speed, making it suitable for real-time applications. Future work will explore the implementation on the mobile platform and the extension to dynamic scenes.

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

# A APPENDIX

## A.1 DETAILS OF SCENE REPRESENTATION AND RENDERING

The scene is modeled as $G = \{g_i\}_{i=1}^N$, with each Gaussian $g_i = (\mu_i \in \mathbb{R}^3, \Sigma_i \in \mathbb{R}^{3\times3}, a_i, \alpha_i \in [0,1])$ defining:

- Mean position $\mu_i$, covariance $\Sigma_i$ (via scale $s_i \in \mathbb{R}^3$, rotation quaternion $q_i \in \mathbb{R}^4$).
- View-dependent color via spherical harmonics (SH) coefficients $a_i$.
- Opacity $\alpha_i$.
- SLAM bookkeeping: observation counts $n_{\text{obs},i}$, maximum screen-space radii $\max_{\text{radii2D},i}$, covisibility identifiers.

The differentiable renderer, transforms Gaussian $i$ for camera pose $T_j = [R_j | t_j] \in \text{SE}(3)$, parameterized by twist $\xi_j \in \mathbb{R}^6$:

$$x_c = R_j\mu_i + t_j, \quad \Sigma_c = R_j\Sigma_i R_j^T, \tag{1}$$

$$J_{\text{proj}}(x_c) = \begin{bmatrix} \frac{f_x}{Z} & 0 & -\frac{f_x X}{Z^2} \\ 0 & \frac{f_y}{Z} & -\frac{f_y Y}{Z^2} \end{bmatrix}, \quad \Sigma_{\text{img}} = J_{\text{proj}}\Sigma_c J_{\text{proj}}^T + \varepsilon I_2, \tag{2}$$

using intrinsics $K = \text{diag}(f_x, f_y)$, principal point $(c_x, c_y)$, and $\varepsilon = 10^{-6}$. Per-pixel weight:

$$w_{i,p} = \exp\left(-\frac{1}{2}(x_p - \mu_i^{2D})^T \Sigma_{\text{img}}^{-1}(x_p - \mu_i^{2D})\right),$$

Table 5: PSNR, SSIM, and LPIPS Metrics for Different SLAM Methods on Replica Dataset. M1 refers to NICE-SLAM Zhu et al. (2022), M2 refers to Vox-Fusion Yang et al. (2022), M3 refers to ESLAM Johari et al. (2023), M4 refers to Point-SLAM Sandström et al. (2023), M5 refers to GS-SLAM Matsuki et al. (2024), M6 refers to SplaTAM Keetha et al. (2024), M7 refers to RTG-SLAM Liu et al. (2024), M8 refers to GLORIE-SLAM Zhang et al. (2024), M9 refers to Photo-SLAM Wang et al. (2024), M10 refers to SplatMAP Hu et al. (2025b). For each method, the first row corresponds to PSNR, the second row to SSIM, and the third row to LPIPS. The best results for each metric are highlighted in bold.

| Input | Method | Room0 | Room1 | Room2 | Office0 | Office1 | Office2 | Office3 |
|---|---|---|---|---|---|---|---|---|
| RGBD | M1 | 22.12 | 22.47 | 24.52 | 27.09 | 30.34 | 19.66 | 22.83 |
| | | 0.76 | 0.76 | 0.81 | 0.84 | 0.91 | 0.77 | 0.81 |
| | | 0.24 | 0.30 | 0.30 | 0.26 | 0.28 | 0.20 | 0.21 |
| | M2 | 22.33 | 22.36 | 22.37 | 27.79 | 29.83 | 20.33 | 23.47 |
| | | 0.79 | 0.79 | 0.81 | 0.84 | 0.91 | 0.78 | 0.83 |
| | | 0.28 | 0.30 | 0.30 | 0.25 | 0.26 | 0.18 | 0.21 |
| | M3 | 25.36 | 27.77 | 29.23 | 29.08 | 32.57 | 28.36 | 32.62 |
| | | 0.91 | 0.93 | 0.94 | 0.97 | 0.97 | 0.96 | 0.98 |
| | | 0.31 | 0.30 | 0.30 | 0.29 | 0.23 | 0.21 | 0.20 |
| | M4 | 29.10 | 31.12 | 31.01 | 35.18 | 38.77 | 35.04 | 32.34 |
| | | 0.97 | 0.98 | 0.98 | **0.99** | **0.99** | **0.99** | **0.99** |
| | | 0.11 | 0.12 | 0.11 | 0.09 | 0.07 | 0.09 | 0.08 |
| | M5 | 31.56 | 32.86 | 35.08 | 37.80 | 41.17 | 39.01 | 33.92 |
| | | 0.97 | 0.97 | 0.97 | **0.99** | **0.99** | **0.99** | **0.99** |
| | | 0.09 | 0.09 | 0.09 | 0.07 | 0.04 | 0.06 | 0.06 |
| | M6 | 32.86 | 33.89 | 35.95 | 38.26 | 41.28 | **39.86** | 32.92 |
| | | 0.97 | 0.97 | 0.97 | **0.99** | 0.98 | **0.99** | **0.99** |
| | | 0.07 | 0.10 | 0.09 | 0.09 | 0.07 | 0.06 | 0.09 |
| | M7 | 34.15 | **34.21** | 35.57 | 37.91 | 41.27 | 38.22 | 35.81 |
| | | 0.979 | **0.981** | 0.981 | **0.99** | **0.99** | 0.98 | 0.98 |
| | | 0.13 | 0.13 | 0.12 | 0.12 | 0.11 | 0.13 | 0.12 |
| MONO | M8 | 30.56 | 30.97 | 28.42 | 31.63 | 32.32 | 31.61 | 32.98 |
| | | 0.96 | 0.97 | 0.96 | 0.97 | 0.98 | 0.98 | 0.98 |
| | | 0.13 | 0.13 | 0.11 | 0.10 | 0.09 | 0.10 | 0.11 |
| | M9 | 29.87 | 29.01 | 29.41 | 32.75 | 33.59 | 31.62 | 34.17 |
| | | 0.87 | 0.91 | 0.91 | 0.95 | 0.96 | 0.94 | 0.96 |
| | | 0.10 | 0.11 | 0.09 | 0.08 | 0.07 | 0.10 | 0.09 |
| | M10 | 35.367 | 31.746 | 38.117 | 42.858 | 42.062 | 35.504 | 39.034 |
| | | **0.98** | 0.95 | **0.99** | **0.99** | **0.99** | **0.99** | **0.99** |
| | | **0.03** | **0.09** | **0.03** | **0.01** | **0.02** | **0.04** | **0.04** |
| | **Ours** | **36.12** | **34.21** | **38.45** | **42.91** | **43.12** | 36.78 | **39.87** |
| | | **0.98** | **0.96** | 0.97 | 0.98 | 0.97 | 0.98 | 0.97 |
| | | **0.03** | **0.09** | **0.03** | **0.01** | **0.02** | **0.04** | **0.04** |

color $c_{i,p} = \text{SH}(a_i, v_p)$, transmittance $T_{i,p} = \prod_{k<i}(1 - \alpha_k w_{k,p})$. Rendered pixel:

$$\hat{C}_p(G, T_j) = \sum_{i \in S_p} w_{i,p} c_{i,p} \alpha_i T_{i,p},$$

where $S_p$ is depth-ordered Kerbl et al. (2023). Outputs include color, depth, gradients $\partial \hat{C}_p / \partial(\xi_j, \mu_i, \Sigma_i, a_i, \alpha_i)$, and masks.

## A.2 DETAILS OF PHOTOMETRIC OPTIMIZATION

Per-pixel residual: $r_{j,p}(G, T_j) = I_{j,p} - \hat{C}_{j,p}(G, T_j)$. Robust loss:

$$\phi(r_{j,p}) = (1 - \lambda_{\text{SSIM}})\rho(|r_{j,p}|) + \lambda_{\text{SSIM}}(1 - \text{SSIM}(I_j, \hat{I}_j)), \quad \lambda_{\text{SSIM}} = 0.85,$$

using Huber surrogate $\rho$. Joint optimization loss:

$$\mathcal{L}(\Theta) = \sum_{j \in W} \sum_{p \in \Omega_j} w_p \phi(r_{j,p}) + \lambda_{\text{reg}} \sum_{i \in G_W} (\|\text{diag}(\Sigma_i)\|_2^2 + |\alpha_i - 0.5|^2),$$

where $\Theta = (\{T_j\}, G_W)$, and the regularizer stabilizes optimization Matsuki et al. (2024); Engel et al. (2017).

### A.3 DETAILS FRONTEND: FRAME-TO-MODEL TRACKING

The frontend estimates the pose of frame $I_t$ with prior $T_{\text{prev}}$, minimizing pose-only:

$$\mathcal{L}_{\text{track}}(T_t) = \sum_{p \in \Omega_t} w_p \phi(r_{t,p}(G, T_t)),$$

with the map $G$ fixed. Update $\xi_t$ using Adam:

$$g_{\xi_t} = -\sum_p w_p \psi_{t,p} \left( \frac{\partial \hat{C}_{t,p}}{\partial \xi_t} \right)^T, \quad \psi_{t,p} = \phi'(r_{j,p}), \quad \xi_t \leftarrow \text{Adam}(\xi_t, g_{\xi_t}, \alpha_{\text{pose}}), \quad T_t \leftarrow \exp(\xi_t) T_t.$$

Insert keyframe if:

$$\|t_t - t_{\text{last}}\|_2 > \tau_t, \quad \text{angle}(R_t R_{\text{last}}^{-1}) > \tau_R, \quad \text{or} \quad \frac{|S_t \cap S_{\text{last}}|}{|S_t \cup S_{\text{last}}|} < \tau_c,$$

with $\tau_t = 0.1 \,\text{m}, \tau_R = 5°, \tau_c = 0.7$. Packages include $I_t$, $T_t$, visible Gaussians.

### A.4 BACKEND: MAPPING STAGE

The mapping stage refines keyframe poses and the Gaussian map through a joint optimizer, the core engine, following the flow: differentiable rendering $\rightarrow$ loss computation $\rightarrow$ joint optimization. For a covisible window $W$ (size $N$), render views using `gaussian_model.py`, compute loss $\mathcal{L}(\Theta)$ (Eq. equation A.2) and backpropagate:

$$g_\theta = -\sum_{j \in W} \sum_{p \in \Omega_j} w_p \psi_{j,p} \left( \frac{\partial \hat{C}_{j,p}}{\partial \theta} \right)^T, \quad \theta \in \Theta.$$

- Update poses using Adam: $\xi_j \leftarrow \text{Adam}(\xi_j, g_{\xi_j}, \alpha_{\text{pose}})$. - Update Gaussians using SLN. - Update bookkeeping $(n_{\text{obs},i}, \max_{\text{radii2D},i})$.

### A.5 STOCHASTIC LOCAL NEWTON (SLN) OPTIMIZER

SLN, applied to Gaussian parameters $\theta_b$ (e.g., $\mu_i, \Sigma_i, a_i, \alpha_i$), approximates the Hessian diagonally:

$$g_b = -\sum_{j \in W} \sum_{p \in \Omega_j(i)} w_p \psi_{j,p} J_{j,p,b}^T, \quad \psi_{j,p} = \phi'(r_{j,p}), \tag{3}$$

$$h_{\text{diag},b} \approx \sum_{j \in W} \sum_{p \in \Omega_j(i)} s_{j,p} (J_{j,p,b} \odot J_{j,p,b}), \quad s_{j,p} \approx \frac{1}{|r_{j,p}| + \epsilon}, \tag{4}$$

using a Huber surrogate for L1/SSIM ($\epsilon = 10^{-3}$). Damped preconditioner:

$$p_b = \text{clamp}(h_{\text{diag},b} + \beta + \varepsilon, p_{\min}, p_{\max}), \quad \Delta\theta_b = -\alpha_{\text{SLN}} \frac{g_b}{p_b},$$

with $\beta = 10^{-4}, \varepsilon = 10^{-6}, p_{\min} = 10^{-6}, p_{\max} = 10^6$. Rotations use expmap Anonymous (2025); Hu et al. (2025a).

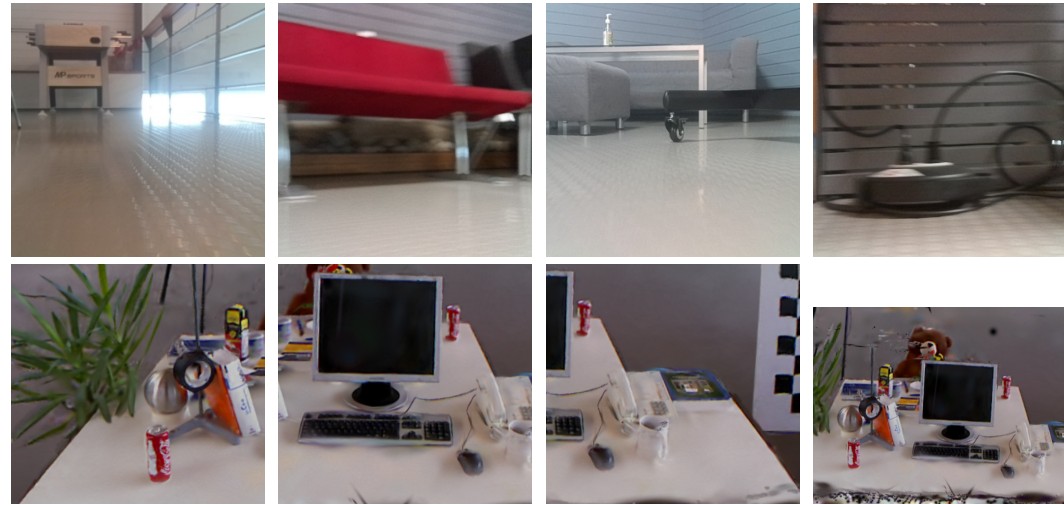

Figure 7: Example visual results of our method. Top row: four samples from our lab, bottom row: four samples from `room1` of Replica. Our approach reconstructs photorealistic scenes with sharp object boundaries and faithful color, even in challenging cases.

### A.6 MAP MAINTENANCE

Densify if:

$$\max \mathrm{radii2D}_i > \tau_{\mathrm{split}} \quad \text{or} \quad \mathrm{grad\_energy}_i = \sum_{j,p \in \Omega(i)} \|J_{j,p,i}^T \psi_{j,p}\|^2 > \tau_{\mathrm{grad}}.$$

Prune if:

$$n_{\mathrm{obs},i} < \tau_{\mathrm{obs}} \quad \text{or} \quad \overline{w_{i,p}} < \tau_{\mathrm{contrib}}.$$

Reset opacity for non-visible Gaussians at cadence $k_{\mathrm{reset}}$. Dynamic objects are supported via pixel-level masks, extending Zhou et al. (2025).

### A.7 GLOBAL CONSISTENCY

Covisibility: $C_{ab} = |S_a \cap S_b|$. Edge if $C_{ab} > \tau_{\mathrm{loop}}$. Pose-graph error:

$$E_{\mathrm{pose}} = \sum_{(a,b) \in \mathcal{E}} \left\| \log \left( (T_a^{-1} T_b) \hat{T}_{ab}^{-1} \right) \right\|_{\Sigma_{ab}^{-1}}^2.$$

### A.8 IMPLEMENTATION DETAILS

### A.9 QUALITATIVE RESULTS2

| Category | Parameter | Value |
|---|---|---|
| Results | save_results | True |
| | save_dir | results |
| | save_trj | True |
| | save_trj_kf_intv | 5 |
| | use_gui | False |
| | eval_rendering | False |
| Dataset | type | tum |
| | sensor_type | depth |
| | pcd_downsample | 128 |
| | pcd_downsample_init | 32 |
| | adaptive_pointsize | True |
| | depth_scale | 5000.0 |
| Training | joint_optimization | True |
| | joint_opt_window_size | 8 |
| | joint_opt_mapping_iters | 25 |
| | global_opt_enabled | True |
| | global_opt_interval | 50 |
| | adaptive_lr_enabled | True |
| | lr_decay_factor | 0.95 |
| | init_itr_num | 1050 |
| Tracking/Mapping | tracking_itr_num | 100 |
| | mapping_itr_num | 150 |
| | gaussian_update_every | 150 |
| | gaussian_th | 0.7 |
| | gaussian_extent | 1.0 |
| | kf_interval | 5 |
| | spherical_harmonics | True |
| Learning Rates | cam_rot_delta | 0.003 |
| | cam_trans_delta | 0.001 |
| | position | 0.00016 |
| | feature | 0.0025 |
| | opacity | 0.05 |
| | scaling | 0.005 |
| 3DGS/Optimization | iterations | 30000 |
| | position_lr_init | 0.00016 |
| | percent_dense | 0.01 |
| | lambda_dssim | 0.2 |
| | densification_interval | 100 |

Table 6: Key parameters used in our SLAM and 3DGS experiments.

