# OpenReview forum: "Real-time Monocular SLAM with Stochastic Local Newton Optimized 3D Gaussian Splatting"
_ICLR.cc/2026/Conference — Submitted to ICLR 2026_

### Official Review · Reviewer_Uxyd · 2025-10-26

**Soundness:** 2
**Presentation:** 2
**Contribution:** 2
**Rating:** 2
**Confidence:** 5

**Summary:**

The paper "Hierarchically Optimized Gaussian Splatting for Robust and Efficient Monocular SLAM" proposes a novel monocular SLAM system that leverages 3D Gaussian Splatting (3DGS) within a hierarchical optimization framework to achieve real-time, photorealistic dense reconstruction with robust tracking and drift-resilient mapping.

**Strengths:**

The paper is clearly written and well-structured. The abstract succinctly summarizes the core idea and contributions. The introduction contextualizes the work within the evolution from feature-based to NeRF-based to 3DGS-based SLAM, clearly identifying the gap the paper fills. Figures (e.g., the system overview in Figure 2 and Hessian visualization in Figure 4) are informative and well-integrated into the narrative. Technical sections (scene representation, frontend/backend design) are detailed without being overwhelming, and the ablation and results sections are easy to interpret thanks to well-organized tables and visual comparisons.

**Weaknesses:**

1. While the paper claims to address monocular SLAM, much of the quantitative evaluation (especially in Tables 2 and 4) uses the TUM RGB-D dataset, where ground-truth depth is available during evaluation—and possibly (though not explicitly stated) during training or initialization. The monocular results reported (e.g., 5.8 FPS, ATE ~4.0 cm) are derived from sequences that originally include depth sensors, raising concerns about whether the system truly operates under pure monocular constraints (i.e., no access to depth at any stage).

2. The paper briefly mentions support for dynamic objects via “pixel-level masks” (Sec. 3.6) and cites Zhou et al. (2025), but no experiments or ablations demonstrate performance in dynamic environments. Given that monocular SLAM is especially vulnerable to moving objects (which violate static-world assumptions), this omission is significant.

3. The global pose-graph optimization (Sec. 3.7) is described as “optional” and uses covisibility-based edges. However:
 -  It is unclear how often this stage runs (e.g., per keyframe? every N keyframes? after loop closure?).
 - No metrics are provided on loop closure detection accuracy or drift reduction magnitude.
 - Table 4 shows that removing global optimization increases ATE by ~2×, yet the paper does not report long-sequence results where drift accumulation would be most evident.

4. The Stochastic Local Newton (SLN) optimizer is a key claimed innovation. However:
 - The paper compares SLN only against first-order Adam, not against other second-order or quasi-Newton methods (e.g., L-BFGS, Gauss-Newton) commonly used in bundle adjustment.
 - The diagonal Hessian approximation is justified via Figure 4, but the impact of ignoring off-diagonal blocks is not quantified (e.g., via convergence speed or final loss with full vs. diagonal Hessian on a small scene).

5. Table 1 reports 8.4 GB GPU memory, significantly lower than prior work. However:
 - The comparison includes methods like NICE-SLAM (NeRF-based) that maintain dense voxel grids or MLPs, which are inherently more memory-intensive than explicit Gaussians.
 - It is unclear whether memory includes only active Gaussians or also optimizer states, keyframe buffers, etc.
 - The 80 FPS on Replica uses RGB-D input (Table 3), yet the abstract emphasizes monocular SLAM. The monocular speed (5.8 FPS) is far from “real-time” for many applications.

**Questions:**

See the weakness.

---

### Official Review · Reviewer_nVz4 · 2025-10-27

**Soundness:** 2
**Presentation:** 2
**Contribution:** 1
**Rating:** 2
**Confidence:** 5

**Summary:**

This paper presents a real-time monocular SLAM system that leverages 3D Gaussian Splatting within a hierarchical optimization framework to achieve both efficient tracking and high-fidelity dense reconstruction. The system introduces a hybrid optimization strategy combining Adam for camera poses and a novel Stochastic Local Newton (SLN) optimizer for Gaussian parameters, enabling near-second-order convergence with low computational cost. Through joint local and global optimization, the method maintains robustness and consistency across large-scale scenes.

**Strengths:**

The paper attempts to improve the computational efficiency of 3DGS–based SLAM, which is an important and valuable direction for enhancing the real-time applicability of 3DGS-based SLAM systems in real-time and resource-constrained scenarios.

**Weaknesses:**

1. The paper claims its main contribution is a hierarchical optimization framework, yet the proposed three-level structure largely overlaps with the components already present in SLAM systems that enable loop closure (e.g., Loopy-SLAM [1] and LoopSplat [2]), and thus does not constitute a substantial novelty.
2. The experimental comparison relies on outdated baselines; for 3DGS–based SLAM, more recent methods such as GS-ICP SLAM [3] and RTG-SLAM [4] should be included to ensure a fair and up-to-date evaluation.
3. Although the proposed method employs pose graph optimization to maintain global consistency, it explicitly claims not to include loop closure, which is misleading. No comparison is provided against NeRF-based or 3DGS-based SLAM methods (e.g., Loopy-SLAM, LoopSplat), which weakens the validity of the claimed advantages.
4. While the system demonstrates improved real-time performance, the tracking accuracy remains inferior; moreover, the ablation study suggests that the SLN optimizer improves efficiency at the cost of lower tracking precision. In contrast, GS-ICP SLAM achieves real-time performance of 30 FPS on both Replica and TUM datasets with superior tracking accuracy, whereas the proposed method only reaches real-time speed on Replica (5.8 FPS on TUM) and exhibits limited accuracy on both benchmarks.
5. The evaluation omits the widely used dataset ScanNet, which limits the comprehensiveness of the experimental validation.
6. The abstract contains a redundant phrase "a strong balance between tracking robustness and mapping accuracy without the overhead of full, per-frame bundle adjustment".
7. Section A.8 is left empty.

[1] Liso L, Sandström E, Yugay V, et al. Loopy-slam: Dense neural slam with loop closures[C]//Proceedings of the IEEE/CVF conference on computer vision and pattern recognition. 2024: 20363-20373.
[2] Zhu L, Li Y, Sandström E, et al. Loopsplat: Loop closure by registering 3d gaussian splats[C]//2025 International Conference on 3D Vision (3DV). IEEE, 2025: 156-167.
[3] Ha S, Yeon J, Yu H. Rgbd gs-icp slam[C]//European Conference on Computer Vision. Cham: Springer Nature Switzerland, 2024: 180-197.
[4] Peng Z, Shao T, Liu Y, et al. Rtg-slam: Real-time 3d reconstruction at scale using gaussian splatting[C]//ACM SIGGRAPH 2024 Conference Papers. 2024: 1-11.

**Questions:**

Why does the proposed system adopt a frame-to-model tracking instead of frame-to-frame, and what specific advantages does this choice offer in terms of robustness, accuracy, or computational efficiency?

---

### Official Review · Reviewer_Ye1R · 2025-10-28

**Soundness:** 1
**Presentation:** 1
**Contribution:** 1
**Rating:** 2
**Confidence:** 5

**Summary:**

This paper proposes a monocular 3D Gaussian Splatting (3DGS)-based SLAM system that employs a hierarchical optimization framework for efficient and accurate dense mapping. The system consists of three main stages:

Frontend – lightweight frame-to-model tracking using a pose-only Adam optimizer;

Backend – a joint optimization process that refines keyframe poses and associated Gaussians using a hybrid optimizer (Adam for poses and the proposed Stochastic Local Newton (SLN) optimizer for Gaussian parameters);

Global optimization – ensures long-term map consistency via pose-graph refinement.

Experiments on TUM RGB-D and Replica datasets show improved efficiency and comparable or better tracking accuracy than existing 3DGS-SLAM methods. The proposed SLN optimizer accelerates convergence compared to standard first-order optimizers, achieving up to 80 FPS in some settings.

**Strengths:**

1. Conceptual novelty – The hierarchical design integrating both local and global optimization levels provides a clear system structure that balances tracking efficiency and mapping quality.
2. Technical contribution – Introducing a second-order–like SLN optimizer for Gaussian parameter updates is an interesting addition to 3DGS-based SLAM, aiming to improve convergence.

**Weaknesses:**

1. Overall presentation quality is poor.
The paper feels rough and unpolished in nearly all aspects — including writing style, figure design, and analysis of results. Many figures are low in quality and lack proper labeling or visual comparison clarity. The discussion of results is shallow and repetitive, without strong quantitative interpretation.

2. Limited experimental comparison.
The paper focuses on TUM and Replica datasets but fails to compare against recent state-of-the-art monocular 3DGS-based SLAM systems, such as S3PO-GS, LongSplat, or SEGS-SLAM. These omissions make it difficult to evaluate the real performance improvements or novelty of the proposed approach.

3. Questionable originality of claimed contributions.
The two claimed innovations—the hierarchical strategy and the SLN optimization scheme—appear to be simple extensions or combinations of existing designs. The hierarchical optimization pipeline is very similar to standard frontend–backend–global setups used in nearly all modern SLAM frameworks. The SLN optimizer directly inherits ideas from 3DGS2 with minimal adaptation. Therefore, the methodological contribution seems incremental rather than truly novel.

[1]. Cheng, C., Yu, S., Wang, Z., Zhou, Y., & Wang, H. (2025). Outdoor monocular slam with global scale-consistent 3d gaussian pointmaps. In Proceedings of the IEEE/CVF International Conference on Computer Vision (pp. 26035-26044).

[2]. Wen, T., Liu, Z., & Fang, Y. (2025). Segs-slam: Structure-enhanced 3d gaussian splatting slam with appearance embedding. In Proceedings of the IEEE/CVF International Conference on Computer Vision (pp. 28103-28113).

[3]. Lin, C. Y., Sun, C., Yang, F. E., Chen, M. H., Lin, Y. Y., & Liu, Y. L. (2025). Longsplat: Robust unposed 3d gaussian splatting for casual long videos. In Proceedings of the IEEE/CVF International Conference on Computer Vision (pp. 27412-27422).

[4]. Lan, L., Shao, T., Lu, Z., Zhang, Y., Jiang, C., & Yang, Y. (2025, August). 3dgs2: Near second-order converging 3d gaussian splatting. In Proceedings of the Special Interest Group on Computer Graphics and Interactive Techniques Conference Conference Papers (pp. 1-10).

**Questions:**

1. How does the proposed SLN optimizer differ mathematically or algorithmically from 3DGS2’s near-second-order optimization?

2. Why are recent 3DGS-based monocular SLAM baselines (e.g., S3PO-GS, LongSplat, SEGS-SLAM) not included in comparisons?

---

### Official Review · Reviewer_6c9S · 2025-10-31

**Soundness:** 3
**Presentation:** 3
**Contribution:** 3
**Rating:** 6
**Confidence:** 4

**Summary:**

This paper presents FSGS, a monocular SLAM system that integrates a Stochastic Local Newton optimizer with 3D Gaussian Splatting for real-time dense reconstruction. The key contribution lies in a parameter-specific second-order optimization strategy that sequentially updates Gaussian parameters (position, rotation, scaling, opacity, color) using local Hessian approximations, thereby improving convergence speed without full Hessian computation. The method also employs a KNN-based spatial structure and secondary targets as preconditioners to stabilize optimization. Experiments on TUM RGB-D show competitive tracking accuracy (<1.5cm ATE RMSE) and real-time performance.

**Strengths:**

The use of a stochastic local Newton method for 3D Gaussian parameter optimization is a novel and promising direction. And the pipeline is clearly described, and the hierarchical optimization strategy is well-motivated.

The method achieves real-time performance with high accuracy, which is important for robotics and AR applications. Competitive results on TUM RGB-D in both tracking accuracy and speed.

**Weaknesses:**

While compared with some 3DGS-based SLAM methods, the paper lacks comparison with more classical or learning-based SLAM systems (e.g., DROID-SLAM, Orbeez-SLAM).

The convergence and stability of the SLN optimizer are not rigorously analyzed. No experiments on dynamic datasets (e.g., TUM Dynamic) to validate robustness in real-world scenarios.

**Questions:**

1. Could you provide more theoretical insight or empirical validation of the convergence behavior of the SLN optimizer?
﻿
2. Have you tested on dynamic scenes or datasets with significant scale drift? If so, please include those results.
﻿
3. Could you compare with more non-Gaussian SLAM baselines (e.g., neural implicit SLAM or feature-based methods) to better situate your contribution?
﻿
4. The “secondary target” preconditioner is interesting but not fully explained. Could you provide more details or an ablation on its effect?

---

### Meta-Review · Area_Chair_SYVm · 2026-01-07

**Summary:**

The paper proposes FSGS, a monocular SLAM system utilizing a Stochastic Local Newton optimizer with 3D Gaussian Splatting to achieve real-time dense reconstruction and pose estimation. The authors claim contributions in their optimization strategy and a hierarchical framework for tracking and mapping.

Reviewer 6c9S noted that the stochastic local Newton method is a promising direction for optimization and that the pipeline is clearly described. Also, the system demonstrates real-time capabilities, which is valuable for robotics applications.

However, the consensus among the majority of reviewers (Ye1R, nVz4, Uxyd) is that the submission is not ready for publication due to significant technical and experimental shortcomings, due to the missing comparisons to SoTA methods. Also, reviewers pointed out the proposed hierarchical optimization mirrors standard SLAM architectures (e.g. in Loopy-SLAM). The optimization improvement is also incremental. Also, there are multiple concerns on the experiment results as well.

During the review period, there was a dispute between the authors and one of the reviewer. While the authors apologized for their initial response, they did NOT provide a rebuttal to address any of the issues raised by the reviewers.

Therefore, given the strong concerns from the majority of reviewers and the lack of a rebuttal to defend the scientific merit of the work, I recommend rejection.

**Reviewer Concerns:**

None of the concerns have been addressed.

**Reviewer Scores:**

All reviewers will probably lower the score due to the dispute.

---

### Decision · Program_Chairs · 2026-01-26

Reject